

# Sexual dimorphism and allometry in the sphecophilous rove beetle *Triacrus dilatus*

Maxwell H. Marlowe, Cheryl A. Murphy and Stylianos Chatzimanolis

Department of Biological and Environmental Sciences, University of Tennessee at Chattanooga, Chattanooga, TN, USA

## ABSTRACT

The rove beetle *Triacrus dilatus* is found in the Atlantic forest of South America and lives in the refuse piles of the paper wasp *Agelaia vicina*. Adults of *T. dilatus* are among the largest rove beetles, frequently measuring over 3 cm, and exhibit remarkable variation in body size. To examine sexual dimorphism and allometric relationships we measured the length of the left mandible, ocular distance and elytra. We were interested in determining if there are quantifiable differences between sexes, if there are major and minor forms within each sex and if males exhibit mandibular allometry. For all variables, a *t*-test was run to determine if there were significant differences between the sexes. Linear regressions were run to examine if there were significant relationships between the different measurements. A heterogeneity of slopes test was used to determine if there were significant differences between males and females. Our results indicated that males had significantly larger mandibles and ocular distances than females, but the overall body length was not significantly different between the sexes. Unlike most insects, both sexes showed positive linear allometric relationships for mandible length and head size (as measured by the ocular distance). We found no evidence of major and minor forms in either sex.

## INTRODUCTION

The order Coleoptera, or beetles, is one of the most speciose lineages of all animals with more than 400,000 species described (*Hammond, 1992*). Static allometry (regression analysis of the size of a structure against body size; *Eberhard, 2009*) has been studied extensively in beetles, due to the presence of exaggerated morphologies (sensu *Emlen & Nijhout, 2000*) in many taxa. These studies first became popular in the families Scarabaeidae and Lucanidae where head and/or thoracic horns and mandibles, respectively, are frequently exaggerated (for reviews of earlier studies and other families see *Eberhard & Gutiérrez, 1991*; *Emlen & Nijhout, 2000*; *Emlen, Hunt & Simmons, 2005*; *Miller & Wheeler, 2005*; *Kawano, 2006*). In recent years, static allometry studies have included many other beetle families, including Anthribidae (prothorax length; *Mattos, Mermudes & Moura, 2014*), Cantharidae (male genitalia; *Bernstein & Bernstein, 2002*), Dytiscidae

Corresponding author
Stylianos Chatzimanolis,
stylianos-chatzimanolis@utc.edu

(body size; *Fairn, Alarie & Schulte-Hostedde, 2007a*), Gyrinidae (body size; *Fairn, Alarie & Schulte-Hostedde, 2007b*), and Leiodidae (mandibular horns; *Miller & Wheeler, 2005*).

The rove beetles (Coleoptera: Staphylinidae) are a hyperdiverse family with more than 60,000 species described (unpublished database maintained by A Newton). *Triacrus dilatus* Nordmann belongs in the subtribe Xanthopygina, a monophyletic lineage of 29 neotropical genera, that includes some of the largest and most colorful of all rove beetles. While little is known about the behavior and natural history of xanthopygine beetles (*Chatzimanolis, 2003*; *Chatzimanolis, 2014a*), *T. dilatus* appears to have a fascinating natural history, occupying the nest refuse piles of the large paper wasp *Agelaia vicina* (de Saussure) in the Atlantic forests of Brazil, Argentina and Paraguay (*Wasmann, 1902*; *Kistner, 1982*). Adults and larvae of *T. dilatus* were seen feeding on fly larvae and breeding on the refuse piles of the paper wasp (*Wasmann, 1902*). While many details on the natural history of *T. dilatus* are still lacking, it is possible that *T. dilatus* exhibits the same behavior as *Quedius (Velleius) dilatatus* (Fabricius), a central European species also associated with paper wasps (*Kistner, 1982*). Both *Q. dilatatus* and *T. dilatus* have subserrate (asymmetrical, looking like marginal teeth-like structures pointing forward) antennae (visible on Fig. 3), which is often characteristic of rove beetles associated with social Hymenoptera (*Schillhammer, 2013*; *Chatzimanolis, 2014b*; *Zhao & Zhou, 2015*). According to zur *Strassen (1957)*, *Q. dilatatus* is able to locate the paper wasps nests by following specific semiochemicals emitted by the wasps, and it is likely that *T. dilatus* can do the same.

While completing a taxonomic review of the species (*Chatzimanolis, in press*), one of the authors (SC) was surprised with the sheer variation in mandible length, head size and overall body length among different specimens of *T. dilatus*, both among males and females. While sexual dimorphism is common in xanthopygine rove beetles (e.g., *Chatzimanolis, 2004*; *Chatzimanolis, 2012*), intraspecific variation is typically not present. In this paper we are interested in examining the sexual dimorphism and allometry in *T. dilatus* by measuring and analyzing the relationships between mandible length, ocular distance (head size) and elytra length (as a surrogate for body length) in both males and females. Since variation in mandible length has been associated with differential reproductive strategies in rove beetles (*Forsyth & Alcock, 1990*; *Hanley, 2001*), we are interested in asking if males exhibit mandibular allometry. Additionally, variation in body length or head size is sometimes indicative of major and minor individuals (*Emlen & Nijhout, 2000*) so we want to test whether there is quantifiable sexual size dimorphism in *T. dilatus* and whether there is dimorphism (major and minor individuals) within each sex.

## MATERIALS AND METHODS

### Specimens

Specimens of *T. dilatus* are rather rare in museum collections, despite their relative large length (22–36 mm). We were able to borrow specimens from the following Natural History Museums, and even though this list in not exhaustive, it includes most specimens of *Triacrus* ever collected (*Chatzimanolis, in press*; numbers next to the acronym indicate

how many specimens were used from each museum): Natural History Museum, London (BMNH, 13), Field Museum (FMNH, 15), Museum of Comparative Zoology (MCZ, 1), Naturhistorisches Museum Wien (MNW, 17), Finnish Museum of Natural History (MZH, 3) and the University of Tennessee at Chattanooga Insect Collection (UTCI, 2). Due to the specialized habitat of these beetles and their rarity, it was impossible to have specimens only from a single locality to control for geographic variation. To reduce any geographic bias we included all specimens that we were able to examine, including both sexes. Photographs (Fig. 1) were taken using a Visionary Digital Passport system with a Canon EOS 40D camera and Canon 50 mm and MP-E 65 mm macro lenses. We took photographs in multiple focal plains and then these images were automontaged using Helicon Focus 6.2.2 (http://www.heliconsoft.com/heliconsoft-products/helicon-focus/) to produce a single fully focused image. To establish a method of reliably referring to particular specimens, all specimens lacking unique identification numbers were given a UTCI barcode with a human readable number sequence. These barcodes do not establish ownership of specimens and they are simply used to associate particular measurements with a specimen.

## Measurements

Measurements were taken using an ocular micrometer on an Olympus ZX61 stereomicroscope. Measurements were first recorded as units of the ocular micrometer scale and were converted later into mm. All measurements, as well as the label information for all specimens, are available in Table S1. We measured the length of the left mandible, the ocular distance (i.e., distance between the eyes) and the length of the elytra. In all cases, measurements (Fig. 1) were made between two clear reference points and beetles were positioned carefully to avoid any bias in measuring and to achieve the same plane of view. The left mandible length was measured from the tip of the mandible to the base of the mandible (mandibular condyle) and this was used as an indication of overall mandibular length. Both left and right mandibles are equal in size. The ocular distance was measured in a straight line between the two eyes. This measure was used as a proxy for the overall head width since the concavity of the lateral borders of head makes it hard to establish other precise points for measurements. The elytra length was measured from the tip of the mesoscutellum to the posterior border of the elytra. We used that measure as an indication of overall body length because rove beetles tend to have telescopic abdomens making it impractical to measure overall body length.

## Data analyses

All analyses were performed in SPSS 21 (IBM Corp., Armonk, New York, USA). Data were tested for normality and homogeneity of variances using the Shapiro–Wilk test and Levene's test for equality of variance, respectively. Frequency distributions for each sex and variable were made to visually confirm these unimodal distributions (Figs. S1–S6).

For each variable measured (left mandible, ocular distance and elytra length), a 2-tailed $t$-test was conducted to determine if there were significant differences between males and females. Linear regressions and heterogeneity of slopes tests were performed for the

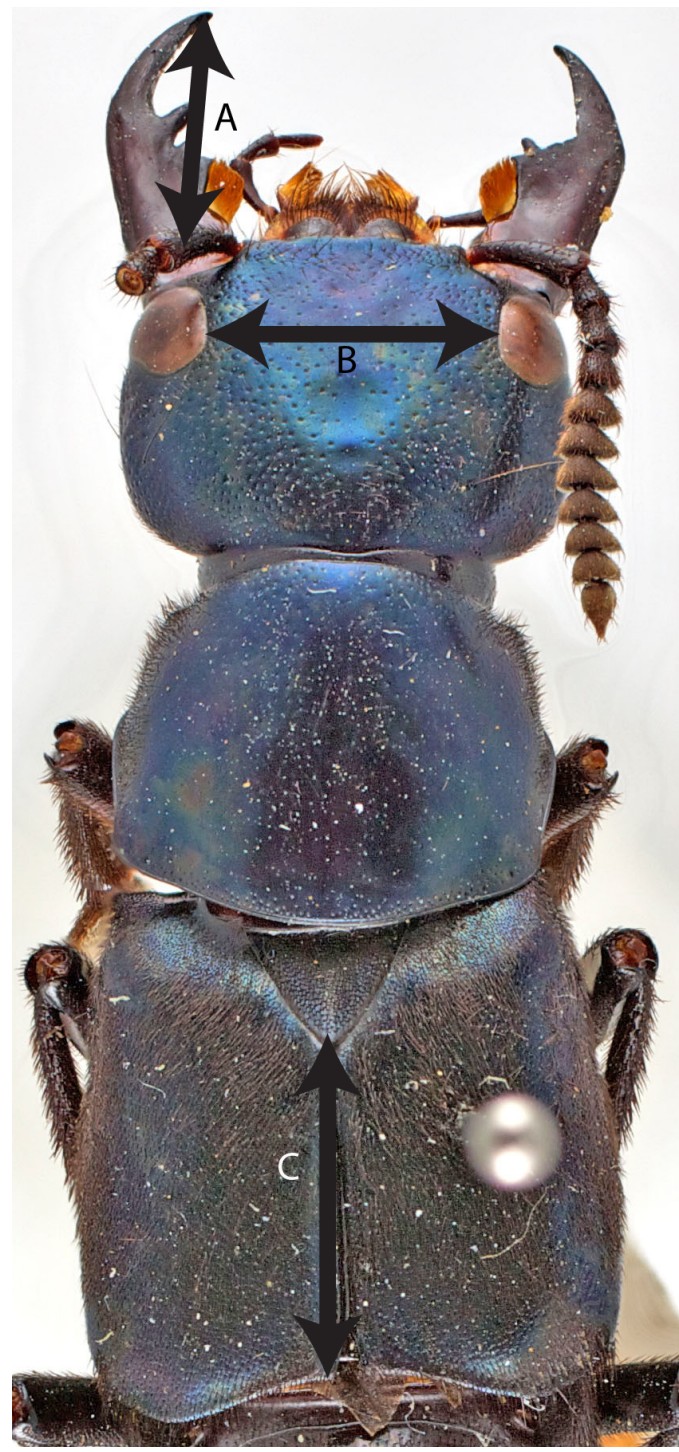

**Figure 1 Dorsal view of *T. dilatus* with measurements.** Dorsal view of *Triacrus dilatus* showing the measurement of left mandible (A), ocular distance (B) and elytra length (C).

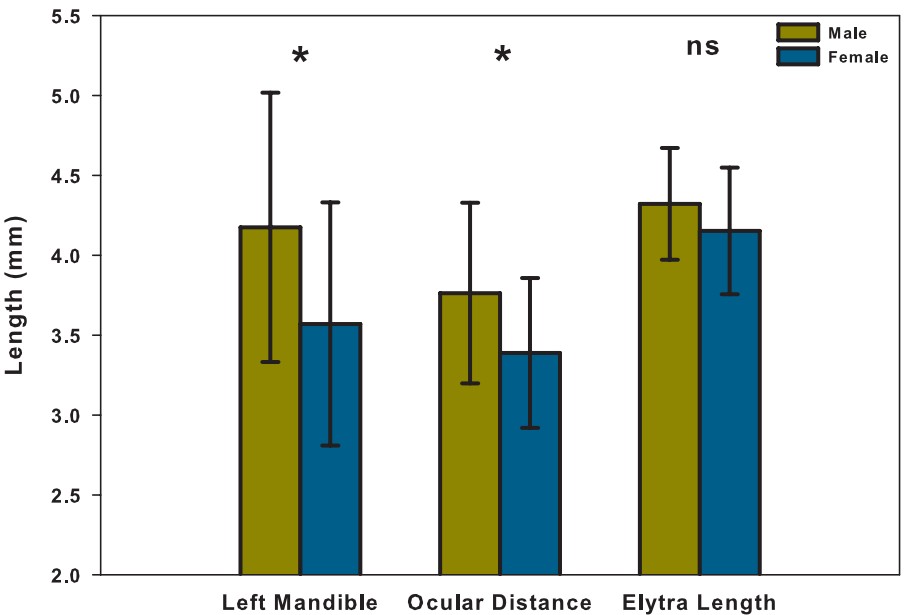

**Figure 2 Means and SD of left mandible, ocular distance and elytra length, both male and female.** Two-tailed $T$-tests were performed to determine significant differences between male and female for each variable: ns, no significant difference; * $p < 0.05$.

following relationships: elytra length and ocular distance, elytra length and left mandible, and ocular distance and left mandible. This allowed us to further examine any differences between males and females by testing for differences in allometry.

In order to determine if there was dimorphism within each sex for each of the measurements, a test for non-linearity was performed by fitting the data to the equation:

$$\ln Y = \alpha_0 + \alpha_1 \ln X + \alpha_2 \ln X^2 + \varepsilon$$

where $\ln Y$ and $\ln X$ are the natural logarithms of the variables of interest; $\alpha_i$ are the regression coefficients; and $\varepsilon$ is the random component (after *Eberhard & Gutiérrez, 1991*; *Hanley, 2001*; *Kotiaho & Tomkins, 2001*). If the coefficient $\alpha_2$ was not significantly different from zero, the relationship was assumed to be linear and therefore, there was no evidence of dimorphism for that particular sex/trait.

## RESULTS

$T$-test analyses indicated that males had significantly larger left mandibles and ocular distances than females (Left Mandible: $t_{49} = -2.65$, $p = 0.011$; Ocular Distance: $t_{49} = -2.51$, $p = 0.015$; Fig. 2). However, the elytra lengths of males and females were not significantly different from each other ($t_{49} = -1.62, p = 0.113$; Fig. 3).

Linear regression analyses showed that every relationship was significant and positive (Table 1 and Figs. 3–5). When comparing the linear regressions between males and females for elytra length vs. left mandible, we found that both the slopes and intercepts for the relationships were not significantly different from one another (Table 2 and Fig. 3). The same was found when comparing males and females for ocular distance vs. left mandible;

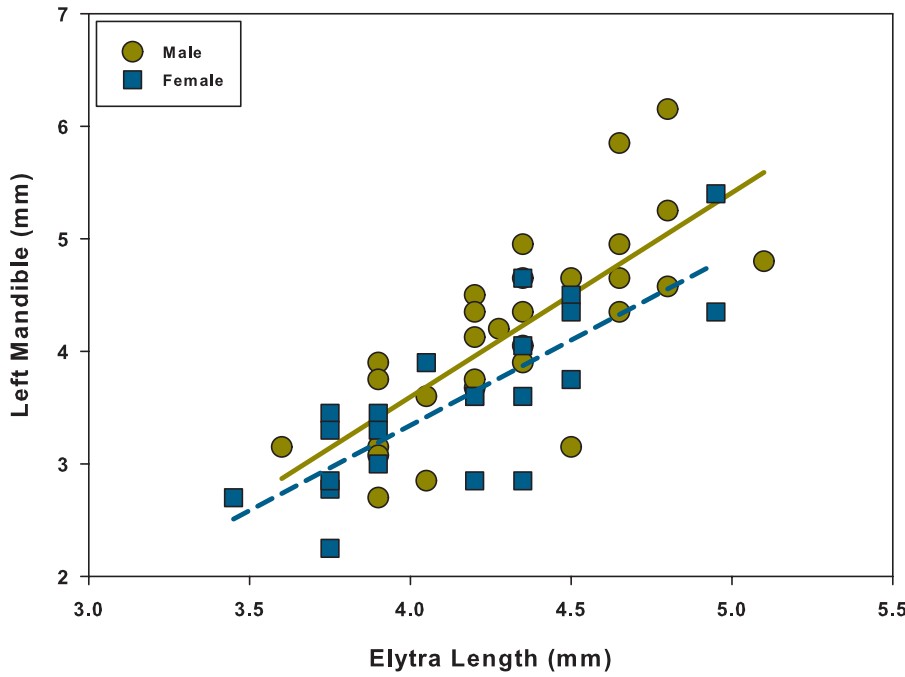

**Figure 3 Scatter plot and linear regression between elytra length and left mandible for males and females.** $N = 29$ males; $N = 22$ females.

**Table 1 Linear regression results for males and females for all three relationships.**

| | Elytra length vs. left mandible | | Elytra length vs. ocular distance | | Ocular distance vs. left mandible | |
|---|---|---|---|---|---|---|
| | **Male** | **Female** | **Male** | **Female** | **Male** | **Female** |
| $R^2$ | 0.57 | 0.62 | 0.81 | 0.87 | 0.53 | 0.68 |
| *F* statistic | 35.25*** | 32.89*** | 111.44*** | 136.67*** | 30.38*** | 42.02*** |
| Intercept | −3.66** | −2.71* | −2.51*** | −1.20** | 0.10ns | −0.96ns |
| Slope | 1.81*** | 1.51*** | 1.45*** | 1.11*** | 1.08*** | 1.34*** |

**Notes.**

$R^2$, correlation coefficient; *F* Statistic, from the regression ANOVA; Intercept and Slope, regression coefficients. Degrees of freedom: 27, males; 22, females; ns, not significantly different from zero.

\* $p < 0.05$.

\*\* $p < 0.01$.

\*\*\* $p < 0.001$.

there were no significant differences between the two (Table 2 and Fig. 4). However, when examining elytra length vs. ocular distance, males had a significantly larger slope than the females (intercepts were not significantly different) (Table 2 and Fig. 5). This indicates that for equal increases in elytra length (or body length), males would have a larger increase in ocular distance than females.

To determine if there was dimorphism within each sex (major and minor forms), we tested for non-linearity. We found that for every relationship, for both males and females, $\alpha_2$ was not significantly different from zero (Table 3 and Figs. 3–5). Therefore,

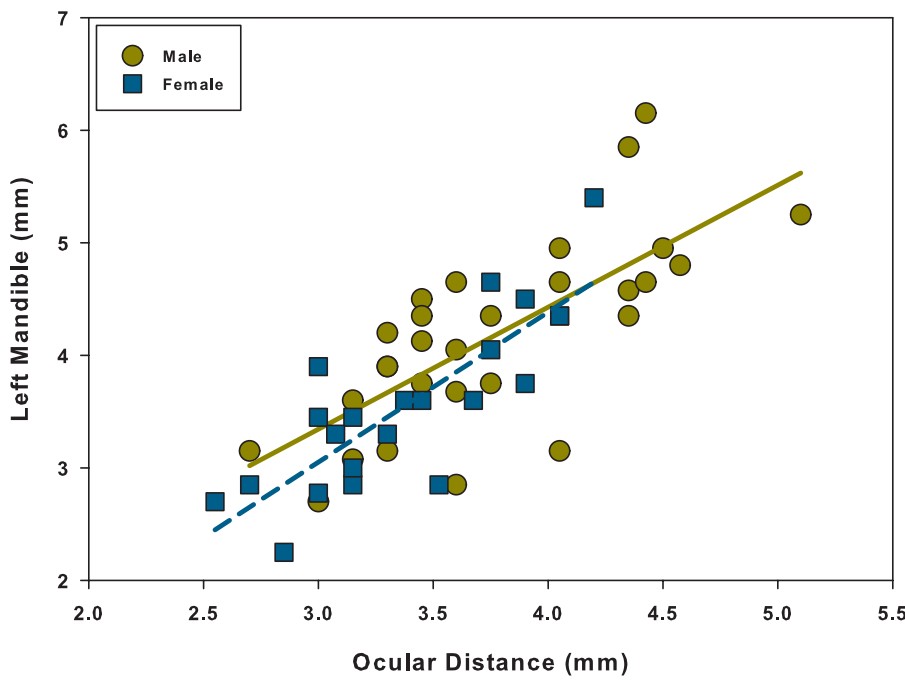

**Figure 4 Scatter plot and linear regression between ocular distance and left mandible for males and females.** $N = 29$ males; $N = 22$ females.

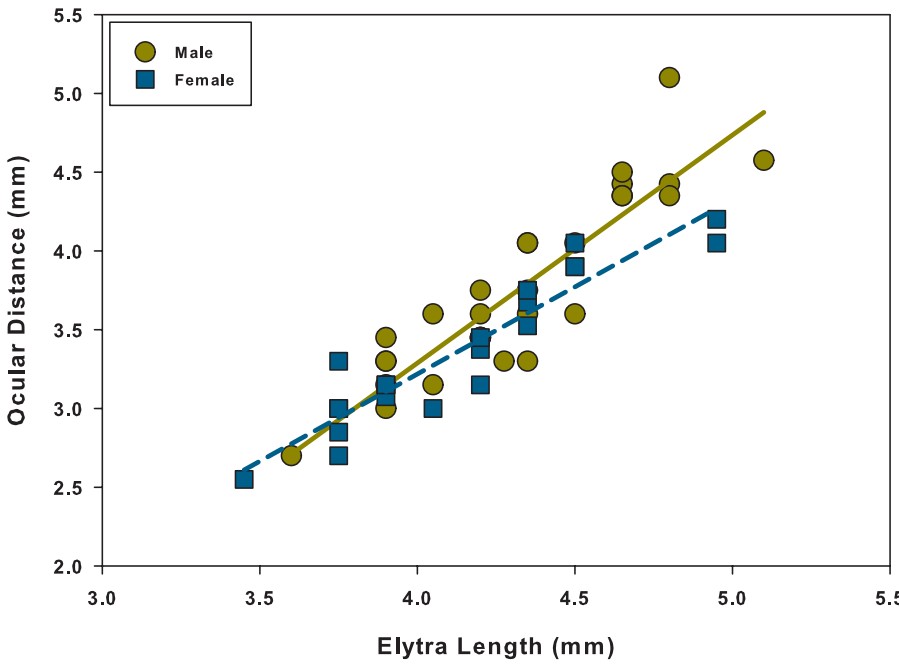

**Figure 5 Scatter plot and linear regression between elytra length and ocular distance for males and females.** $N = 29$ males; $N = 22$ females.

**Table 2 Heterogeneity of slopes test results comparing linear regressions between male and females for each relationship.**

| | Elytra length vs. left mandible | | Elytra length vs. ocular distance | | Ocular distance vs. left mandible | |
|---|---|---|---|---|---|---|
| | $F_{1,47}$ | $p$-value | $F_{1,47}$ | $p$-value | $F_{1,47}$ | $p$-value |
| Intercept | 0.30 | 0.587 | 3.22 | 0.079 | 0.94 | 0.336 |
| Slope | 0.54 | 0.466 | 4.07 | 0.049[*] | 0.68 | 0.413 |

Notes.
Intercept and Slope, regression coefficients.
[*] $p < 0.05$.

**Table 3 Levels of significance for coefficient $\alpha_2$ in Model 1.**

| | Elytra length vs. left mandible | | Elytra length vs. ocular distance | | Ocular distance vs. left mandible | |
|---|---|---|---|---|---|---|
| | Male | Female | Male | Female | Male | Female |
| $\alpha_2$ | −1.27 | 1.24 | 0.22 | −0.87 | −0.20 | 1.64 |
| $t$ | −0.38 | 0.41 | 0.14 | −0.77 | −0.19 | 1.16 |
| $p$-value | 0.706 | 0.687 | 0.891 | 0.449 | 0.852 | 0.262 |

Notes.
Degrees of freedom: Males, 2, 26; Females, 2, 19. Significance at $\alpha = 0.05$.

each relationship was linear in nature and there was no indication of dimorphism within each sex for any of the relationships. Because we did not find dimorphism within each sex, no further analyses were performed to determine a switch point for the variable of interest. Because the data had a normal distribution, equal variances and linear relationships (Tables S2–S3), data are shown without any transformations (i.e., natural log transformation).

## DISCUSSION

Our results showed that there are significant differences in mandibular length and head size (ocular distance) between males and females, with males having larger mandibles and heads. It is possible that mandible size is not entirely independent from the size of the head, because larger mandibles will probably require more room for the muscles associated with them. There are no behavioral data on how male (or female) *T. dilatus* use their mandibles besides capturing food. It is unlikely that they use them to interact with the paper wasps *A. vicina*, whose nest refuse piles they are occupying, because *A. vicina* is a well-studied organism (e.g., *Zucchi et al., 1995*; *Sakagami et al., 1996*; *Baio et al., 1998*; *Mancini et al., 2006*; *De Oliveira, Noll & Wenzel, 2010*; *Moretti et al., 2011*; *Souza et al., 2013*) and there are no reported interactions.

Despite the sheer diversity of rove beetles, there have been just a handful of studies in this group dealing with sexual dimorphism using a statistical approach (*Forsyth & Alcock, 1990*; *Thayer, 1992*; *Hanley, 2001*). *Thayer (1992)* showed that there was

sexual wing dimorphism in *Omalium flavidum* Hamilton, while the other two studies examined mandibular allometry in males of *Leistrotrophus versicolor* (Gravenhorst) (*Forsyth & Alcock, 1990*) or in the genus *Oxyporus* (*Hanley, 2001*). Both the *L. versicolor* and the *Oxyporus* studies revealed the presence of major and minor males (as evident by mandibular allometry), something that we did not detect in *T. dilatus*. Both of these studies (*Forsyth & Alcock, 1990*; *Hanley, 2001*) attributed the presence of major and minor males to different reproductive strategies among males living in ephemeral habitats (vertebrate dung for *L. versicolor* and fungi for *Oxyporus*). On the other hand, the habitat of *T. dilatus*, appears to be less ephemeral with the nests of *Agelaia vicina* being the largest of all wasps and bees (*Zucchi et al., 1995*) and lasting for more than six months (*De Oliveira, Noll & Wenzel, 2010*). While we do not have any direct observations of *T. dilatus*, based on the mandibular analyses results alone, it appears that males of *T. dilatus* do not operate in a similar fashion as *L. versicolor* and *Oxyporus*. Perhaps *T. dilatus* do not have male differential reproductive strategies (at least with respect to morphology), although they could be using their mandibles for competition and resource allocation.

The majority of insect allometric studies have identified exaggerated morphologies in males (review by *Emlen & Nijhout, 2000*; *Bernstein & Bernstein, 2002*; *Emlen, Hunt & Simmons, 2005*; *Tomkins, Kotiaho & LeBas, 2005*; *Mattos, Mermudes & Moura, 2014*). Examples of positive static allometry in females are not common in the literature (but see *Kelly, 2014*). However, our results indicate a positive allometric relationship regarding mandibular size and ocular distance in both males and females. In fact, all slopes were larger than 1.0 and linear for every relationship examined (for both sexes) without significant differences between the sexes. The only significant difference between the sexes was that males would have a higher increase in head size for equal increase in body length. That result could imply that males tend to allocate more resources in building a larger head, perhaps for male–male competition, but we have no empirical evidence to support this claim. In addition, despite the presence of large and small individuals, we found no statistical evidence for dimorphism within each sex since our slopes were linear and not sigmoid or completely broken as expected in taxa with major and minor individuals (*Emlen & Nijhout, 2000*; *Hanley, 2001*).

*Teder & Tammaru (2005)* indicated that in 80% of all examined insect species the females were the larger sex. In our study, while there is a trend for larger males than females (as measured by elytra length), there is no significant sexual size dimorphism in *T. dilatus*. *Teder & Tammaru (2005)* and *Kawano (2006)* found out that the mean body length was larger in sexually dimorphic species. However, it appears that xanthopygine rove beetles (where *T. dilatus* belongs) do not follow this pattern. Examination of the largest (in terms of length) genera in the subtribe (*Triacrus* Nordmann, *Trigonopselaphus* Gemminger and Harold, *Terataki* Chatzimanolis and *Elmas* Blackwelder) showed that while there is sexual dimorphism in all these genera, there were no significance differences in size between males and females (this study; S Chatzimanolis, 2015, unpublished data). Further examination of all genera in the subtribe Xanthopygina revealed that while sexual dimorphisms are common, there were no significance differences in size between

males and females (S Chatzimanolis, 2015, unpublished data). Due to the lack of data regarding this hyperdiverse family, it is unclear if this is a general trend in rove beetles or if xanthopygines are the exception.

Even though sexual size dimorphism is not widespread in xanthopygine rove beetles, sexual dimorphism (or presence of secondary sexual traits) is common in many different genera. In xanthopygine rove beetles, males have modified abdominal genital sternites (e.g., *Ashe & Chatzimanolis, 2003*; *Chatzimanolis, 2004*; *Chatzimanolis, 2008*; *Chatzimanolis, 2015*), presence of a glandular porose structure on abdominal sternite VII (*Chatzimanolis, 2013*; *Chatzimanolis, 2015*) or modified antennae (*Chatzimanolis, 2012*), features that are all missing or not modified in females. *Chatzimanolis (2005)* examined in a phylogenetic context the evolution of secondary sexual structures, and specifically the modification on abdominal sternite VIII (the last abdominal ventral segment before the genital segment), among different species of *Nordus* Blackwelder. He found that while there was no variation within species, the same structure seemed to have changed multiple times among different species. While *Triacrus* is a monotypic genus and we cannot compare variation among different species, it is remarkable to report such variation within species. The results reported here for the genus *Triacrus* are at odds with the pattern observed in *Nordus* (no intraspecific variation) and other Xanthopygina genera (e.g., *Plociopterus* Kraatz; S Chatzimanolis, pers. obs., 2015), It is possible, however, that the lack of variation within species for the genera *Nordus* and *Plociopterus* is due to insufficient observations and not a true pattern.

## CONCLUSIONS

Our analyses documented positive mandibular and ocular distance allometry in both males and females. We also uncovered significant differences between males and females regarding mandibular length and ocular distance (head size). While males had larger mandibles and head, the average body length between males and females was not significantly different; therefore no significant sexual size dimorphism was detected. All allometric relationships were linear and we were unable to find discrete major or minor individuals in either sex.

## ACKNOWLEDGEMENTS

We thank the curators and collections managers of the collections listed in the Materials and Methods section for the loan of specimens. Lu Musetti and Jim Carpenter helped us clarify the correct name of the paper wasp.

### Funding

Partial financial support was provided by a Brock Honors scholarship from UTC (to MM) and a Research and Creative Activity Award from the College of Arts and Sciences, UTC (to SC). The funders had no role in study design, data collection and analysis, decision to publish, or preparation of the manuscript.

## Grant Disclosures

The following grant information was disclosed by the authors:

UTC.

College of Arts and Sciences.

## Competing Interests

The authors declare there are no competing interests.

## Author Contributions

- Maxwell H. Marlowe performed the experiments, wrote the paper, prepared figures and/or tables, reviewed drafts of the paper.
- Cheryl A. Murphy analyzed the data, contributed reagents/materials/analysis tools, wrote the paper, prepared figures and/or tables, reviewed drafts of the paper.
- Stylianos Chatzimanolis conceived and designed the experiments, contributed reagents/materials/analysis tools, wrote the paper, prepared figures and/or tables, reviewed drafts of the paper.

## Supplemental Information

Supplemental information for this article can be found online at http://dx.doi.org/10.7717/peerj.1123#supplemental-information.

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
