# Peer review of "Sexual dimorphism and allometry in the sphecophilous rove beetle Triacrus dilatus"

_PeerJ, doi:10.7717/peerj.1123_

## Round 0.1 · original submission · Major Revisions

The reviewers have identified two major issues concerning your contribution as well as numerous lesser concerns. One reviewer recommended rejection and the other major revision. The two major issues are (1) whether your mandible length measurements are reliable due to variation in opening angle and the plane of view and (2) whether you can meet your objectives by combining specimens from a wide range of locations, given the potential for variation in allometric patterns among locations. If you cannot satisfactorily address these two questions, the manuscript will need to be rejected. If you can address these two questions, then the required changes would also include the many other suggestions and queries of the reviewers. Therefore, I have decided to request major revisions to provide an opportunity for you to respond in case there is some misunderstanding.

Reviewer 1 ·

Basic reporting

- Line 27: I think that ‘between the sexes’ is not the right phrase here.
- Line 45: lineages; add a comma after ‘beetles’.
- Line 46: please explain what static allometry is.
- Line 59: is Newton an author or the name of a database (or something else)? Please clarify.
- Line 62: ‘a’ fascinating natural history
- Lines 59, 64, 68: please provide the taxonomic authors in a uniform style. Also, according to the instructions to authors of PeerJ, authors of zoological names should consist of initials plus full surnames.
- Line 65-66: is this your own observation? Or can you provide a reference?
- Line 72: why do you think this is likely?
- Line 75: one of the authors.
- Lines 80-82: please explain in the introduction why you want to investigate these research question. Is there a reason (based on their behaviour, life history, knowledge on related species,…) why you expect that sexual size dimorphism, intrasexual dimorphism or mandibular allometry will be present in this species? Further, please add to the introduction why are you specifically interested in the three parameters that you measured (mandible length, ocular distance and elytra length).
- Line 89: in museum collections? Please specify. Add a comma after ‘collections’.
- Line 96: maybe ‘reduce’ is more appropriate than ‘avoid’?
- Line 97-99 and further on: always give the company name, city and country of the manufacturer of the equipment and software that you used.
- Line 130: ‘the’ data
- Line 133: please specify whether you performed a one-tailed or a two-tailed t-test.
- Line 138: please clarify how the above tests determine differences in allometry.
- Line 145: ‘… lnY and lnX are the natural logarithms of…’
- Line 175-176: this seems to be a repetition of lines 129-130, please clarify what new result you report here or remove the sentence.
- Line 181: there ‘are’ significant differences: please use the present tense when you refer to your own results.
- Line 182: heads
- Line 184: will ‘probably’ require
- Line 201: appears
- Line 200-203: please check the interpunction of this sentence
- Line 213: please rephrase ‘indicated was that’
- Line 216: in my opinion, the size difference between 1A and 1B is not striking. Please rephrase this and/or quantify the difference.
- Line 221: there ‘is’ a trend
- Line 222: what is not surprising? I don’t understand the link between your findings (line 221-222) and those of Teder and Tammaro (line 222-223).
- Line 223-226: this is not a correct summary of Rensch’s law, which states that within lineages, sexual size dimorphism decreases with body size when females are the largest sex, and that sexual size dimorphism increases with body size when males are the largest sex (e.g. doi: 10.1073/pnas.0404503101). Also, I don’t think you can interpret your results in light of this rule, since it applies to lineages, while you studied only one species.
- Line 233 and 235: significant differences
- Lines 226-238: It is not clear to me what the line of thought is in this paragraph. Please consider whether you can rewrite this to make it more clear.
- Figure 1: please clean the background of this figure (especially when printed in greyscale, lighter and darker spots are visible).
- Figure 2: please add numbers or letters to designate the three parameters.
- Figure 3: I think it would be more useful to show the standard deviation, instead of the standard error, because this shows the amount of variation between your specimens.
- Table 1: please explain the reported variables (R², F, Intercept & slope) in the caption.
- Table 2: please use stars to indicate significant values, for uniformity with the other tables and figures.

Experimental design

- Line 91: how do you know that these include most specimens ever collected?
- Line 87: please specify the number of specimens.
- Line 99: what is the purpose of the automontaging in Helicon Focus?
- Line 114-115: the length of the mandible that you measure, depends on the opening angle of the mandible. Did you control this factor? Also, your measurement depends on the position of the mandibles, relative to the microscope. For example, in figure 1, the mandibles of specimen A seem to lie in a plane that is perpendicular to the sight line (as it should be), while those of specimen B seem to lie in an oblique plane. Also the fact that a larger part of the neck is visible in 1B, shows that the head is bent downwards. As a result, you will measure a shorter length than the actual mandible length.
- Line 116: did you test whether the left and the right jaw are equal in size? Often, this is not the case in insect mandibles (e.g. to facilitate cutting and shredding).
- Line 120: I understand that it is impossible to measure the total body length in a repeatable manner because of the telescopic abdomen. However, I think it would be more relevant to measure the total length of the rest of the body (head length + thorax length), instead of only measuring the elytra length.

Validity of the findings

- Line 196-198: is it possible that minor and major males are present in T. dilatus, but that you could not detect it due to the small sample size?
- Line 209-211: This sentence is ambiguous: do you compare your results to those species that have males with exaggerated morphologies, or do you compare them to the statement that in most insect species, males have exaggerated morphologies? I am not sure that the latter statement is correct. It may for example be a side effect of the preference of investigators for species with exaggerated morphologies (who may ignore less impressive species).
- Line 249-251: why is the observation of Chatzimanolis (2005, that there is a large variation of a secondary sexual structure between species, but no variation within a species) in contrast to your results? You investigated only one species, so this conclusion seems too far-reaching to me. Further, I would call your results in contrast to the published data, instead of the opposite way around.
- Please add a conclusion section or concluding paragraph to the discussion section in which you briefly summarize your main findings.

Reviewer 2 ·

Basic reporting

The taxonomic breadth of the paper needs to be widened considerably. Right from the manuscript’s first sentence it is all about beetles and little else. A considerable amount of work has been done in insects with respect to describing the scaling relationships among various traits.

The Introduction starts too specific. I think it would be best to nest your research questions within a conceptually broader context (e.g. factors underlying the evolution of sexual dimorphism in size and allometry). After all, you conclude the Introduction with some questions/predictions: “We ask the following questions: Is there quantifiable sexual size dimorphism in T. dilatus? Is there dimorphism (major and minor individuals) within each sex? Do males exhibit mandibular allometry?” These predictions come from somewhere, correct? You just didn’t fabricate them de novo. It is important to explain to the reader how you came to make these predictions.

I think you should represent the different museums on each scatterplot (perhaps with a letter in each datum). I think it important for readers to know that, for example, one museum contributed only 2 data points.

Experimental design

You say “impossible to have specimens only from a single locality to control for geographic variation” I think this is really important. We know that there are strong inter-population differences in scaling in a number of species. So I'm not sure how you would be able to generalize for the species when there could be dramatic scaling differences among populations in scaling.

Coefficients of variation would be helpful to report as would a table with all summary stats for each trait for each sex.

Why did you use model I regression rather than model II? A mountain of literature has been published on why this approach is superior for describing allometric scaling relationships versus ordinary least squares.

Why were photographs taken? You took measurements from the specimens under a microscope, correct?

Did you ensure that each trait was parallel with the microscope's lens when taking measurements? Your photos look like the jaws tilt downward.

Validity of the findings

This manuscript examines sexual dimorphism in scaling relationships among various traits in the tropical rove beetle Triacrus dilates. The authors collected data on 51 individuals from 6 museums. I very much appreciate the goals of the manuscript; however, I am concerned with trying to make species-level generalizations about scaling patterns by using individuals from widely dispersed localities. This is particularly troublesome given our considerable knowledge of how populations can differ in their scaling relationships. For example, Tomkins work on earwigs in the UK showed tremendous variation in scaling patterns with some populations showing male dimorphisms while others do not.

Additional comments

Page 3
“subserrate antennae” : This type of antenna is specialized for detecting semiochemicals? With this, you are assuming a lot of ento-anatomical knowledge on the part of the reader.
Page 4

“relative large size” : Can you give a body length range here?

“Natural History Museums”: These words do not need to be capitalized. Can you put the sample sizes next to each museum.

“size” : Do you mean length?

“automontaged”: What did this do?

Page 5

Manufacturer info for SPSS 21

“gender”: Replace “gender” with “sex” throughout the text.

Page 6

“size”: Do you mean length?

“differential reproductive strategies”: Perhaps not with respect to morphology but maybe males differ in their mating behaviour.

Page 7

You say “ thus far” and cite a paper published in 2000 as support. There have been many studies published on "exaggerated morphologies" since 2000, please cite some of them. Also, do you mean in beetles only, because there is a large literature on other insects/animals. I strongly recommend you broaden the scope of the paper.

“The only significant difference indicated was that for equal increase in body 214 length males would have a higher increase in head size.” : Re-write this sentence, it is not clear.

“larger males”: Or maybe there was a trend for people to collect larger males for museums. Biased sampling is large problem when using museum specimens to make broad claims.

“insects and beetles”: Aren't beetles insects?

Page 8

“abdominal sternite VIII” : What does this trait look like and do? Again, you assuming considerable ento-anatomincal knowledge on the part of the reader.

---

## Round 0.2 · accepted · Accept

I feel that you have successfully addressed the concerns of the reviewers regarding measurements and drawing conclusions from specimens originating in multiple populations. Most of the other suggestions were appropriately addressed and I am therefore pleased to accept your manuscript without subjecting it to another round of reviews.

Regarding Model I and II regressions, I believe that your response is rather specific to your particular interests. Although I am not particularly statistically sophisticated, my understanding is the Model II (reduced major axis) regressions are appropriate when there is error in both the X and Y variable (as in your case). However, in this situation, it seems unlikely to make a major difference in your results, so you can either leave the regressions as you have them or check into the issue with a statistician and make appropriate changes before finalizing the manuscript.

With regard to the capitalization of Natural History Museums (L96), I agree with the reviewer. When you refer to museums (or universities) in general, the word is not capitalized.